# Ageing Organizations: Reviewing the Literature and Making a Few Recommendations for Human Resource Management

Alfredo Salomão Filho [1,*], Tanja Tillmanns [1] and Trudy Corrigan [2]

1   Institut für Lern-Innovation, Friedrich-Alexander-Universität Erlangen-Nürnberg, 91054 Erlangen, Germany; tanja.tillmanns@ili.fau.de
2   School of Policy & Practice, Dublin City University, D09 YT18 Dublin, Ireland; trudy.corrigan@dcu.ie
*   Correspondence: alfredo.salomao@ili.fau.de

**Abstract:** In an ageing society, organizations must consider the inclusion and effective management of older professionals if they wish to remain competitive. Besides having harmful and far-reaching effects on people's health, age discrimination in the workplace leads to absenteeism, lower levels of productivity, and early retirement. Within such a context, this problematic organizational feature of most Western, now ageing, organizations and labour markets starts gaining more relevance. However, to date there has been little discussion, research, or policy development to ensure that older employees' skills and capabilities are optimally put to use by organizations without the occurrence of ageism. We present the results of a systematic literature review based on 30 studies in the context of ageing organizations to make recommendations for human resource management. We suggest an approach to effectively manage intergenerational organizations, reducing the occurrence of age discrimination and its related consequences, as well as to achieve high employee engagement. Our review indicates that a strong ethics framework that is continuously followed, disseminated, and updated by management, together with a combination of efforts from all stakeholders, may accomplish the best results towards a culture that respects and values people of all ages at work, positively impacting on productivity.

**Keywords:** human resources management; ageing organizations; older employees; organizational policy; recruitment and training; intergenerational; ageism

## 1. Introduction

It might be the case that an individual is considered too old to excel as a world-class pianist and simultaneously considered too young to coach a professional soccer team, if they are 18 years old [1]. Although age is a concept influenced by society [1], ageism represents a troubling "social disease" [2]. Unfavourable opinions regarding elderly individuals are prevalent in our culture, and this phenomenon has persisted over time. Even the Stoic philosopher Seneca (4 BCE–65 CE) viewed old age as a form of illness, while Juvenal (55–127 CE) used derogatory language towards people in their seventies who were dealing with health issues. In more modern times, age has developed into a defining characteristic both for individuals and within the broader context of everyday life, especially within industrialized Western societies that have aged significantly [3].

The 1948 Universal Declaration of Human Rights explicitly asserts the right for everyone to have fair employment opportunities, and the UN 2030 Agenda's Goals 3, 4, and 8 concentrate on fostering an inclusive work environment [4,5]. Globally, ageism is readily apparent in organizations. Older professionals encounter notable discrimination due to age stereotypes, such as the association between older age and inefficacy, inflexibility, and less motivation to learn new skills, especially in the digital realm [1,4,5]. This form of discrimination not only fosters a toxic workplace atmosphere that directly impacts employees' well-being, but also produces adverse effects, such as reducing total productivity and

compelling professionals to retire prematurely [1,6–8]. From a more positive perspective, however, older professionals can be seen as reliable, experienced, socially skilled and hard-working [1], giving space for organizations to capitalize on those strengths via job crafting and effective management.

Given the contemporary trend of increased longevity and healthier living, combined with the growing concept of extended working life, this article maps current management challenges in ageing workplaces. To effectively address this issue, it is imperative to gain a deeper understanding of the intricate connection between productivity and age-based bias, requiring action not only from private enterprises but also from governmental bodies and civil society [1,9].

This work consists of a systematic literature review of 30 selected studies within the context of ageing organizations, encompassing the management of older professionals as well as their recruitment and training issues, followed by intergenerational relations and ageism in organizations. The analysis of the selected studies generated a series of evidence-based recommendations towards a more effective management of ageing organizations, reducing prejudice, stereotypes and discrimination against older employees.

The recommended plan of action is to redesign and enact organizational policy and regulations to accommodate the needs of older professionals, supporting and encouraging their work performance, intrinsic motivation, and physical and psychological health. Organizations are advised to engage with older employees, promoting organizational intergenerational "oneness" and eliminating the "age norming of jobs" [10,11]. The review indicates that a combination of efforts from all stakeholders may accomplish the best results towards a culture that respects and values older adults at work, which in turn sets the conditions for enhanced organizational performance.

## 2. Materials and Methods

This research constitutes an independent systematic literature review [12] with the objective of delving into the literature [13] on ageing organizations and their associated management challenges. To achieve this, a descriptive textual narrative synthesis methodology was employed [14]. This descriptive review method involves assessing the present status of the literature and concentrating on particular subject areas [14]. Therefore, 4 thematic areas concerning older professionals were predefined to orient both our literature search and the writing up of the results, as follows:

1. Managing older professionals;
2. Recruitment and training of older professionals;
3. Intergenerational relations in organizations;
4. Ageism in the workplace.

The above-mentioned thematic areas aid in structuring the chosen literature, as the textual narrative synthesis involves utilizing a uniform data extraction template that centres on various aspects of the literature, including its outcomes and context, thus forming the core of the review [15,16]. Because of the standardized approach taken in our review, we incorporated quantitative and qualitative studies, along with theoretical and empirical research, related to each subject area. To ensure reliability, we followed the PRISMA guidelines [17]. The PRISMA flow diagram is displayed below (Figure 1).

In what follows, the entire review process is explained in detail in relation to the inclusion criteria and literature search and evaluation.

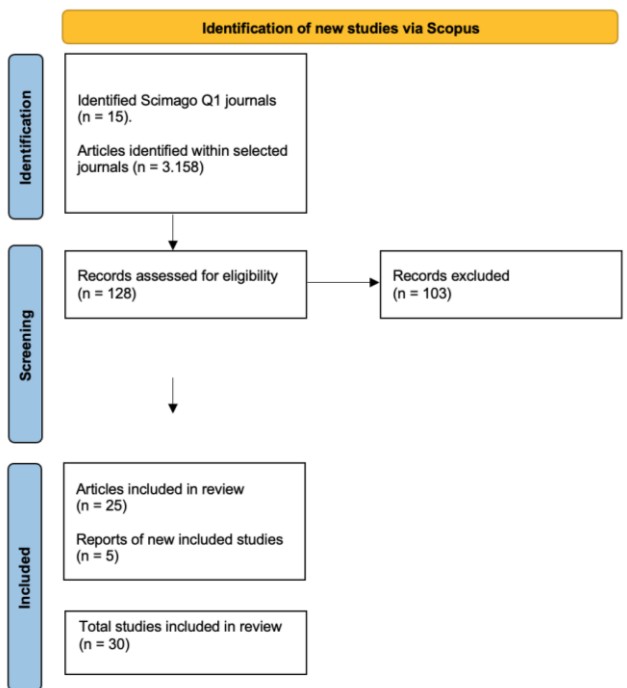

**Figure 1.** PRISMA 2020 flow diagram (adapted from Page et al., 2021 [17]).

### 2.1. Inclusion Criteria

This systematic literature review comprises 30 studies, including peer-reviewed articles (both theoretical and empirical), books and reports published within the timeframe of the two years spanning from 2020 to 1 July 2022. The selected time span ensures that the review is founded on the most current knowledge available, incorporating recent policy and practice developments.

Encompassing a range of fields from social to business studies, the search exclusively considered peer-reviewed articles ranked within Scimago Quartile 1 (*n* = 25). In addition to the journal-sourced literature, the review also includes one book [6] and three book chapters originating from a curated book [18–20]. Furthermore, the review took into account the 2021 UN World Health Organization's Global Report on Ageism [1], a pivotal resource: it diversifies the publication types in order to enhance the review's robustness, while adhering to the established date criteria.

Nonetheless, there are certain constraints associated with this study. The review only included publications in English and did not assess the methodological rigor of each scientific article; the conclusions of the studies were taken as presented. The fact that the selected articles hold prominent positions in Scimago, however, indicates their reliability and credibility.

### 2.2. Literature Search and Evaluation

The study's framework revolved around specific themes in the investigation of ageing organizations, which were subsequently refined. Following the application of the chosen criteria for inclusion, the relevant literature was surveyed. This process was continued via the screening and analysis of the material, culminating in the compilation of results and the present work.

The search for journals in Scopus was carried out using keywords, such as "ageing", "aging", "organizational psychology", "organizational behavior", and "human resources". The preliminary relevance of the manuscripts was assessed based on their titles within each journal. After identifying the journals, keywords such as "older workers", "employment", "ageism", "agism", "senior", and the combination "intergenerational learning AND work" were employed to locate pertinent articles. A two-step process was used for the selection: titles were screened

initially, and then relevant abstracts were evaluated to determine which articles would be included in the review. A total of 3158 articles from 15 journals underwent this process. During the screening, 128 peer-reviewed articles were examined for eligibility but were ultimately excluded; they were more closely related to social policy on retirement, end-of-career opportunities, and the health and well-being of older workers.

For this review, the relevant literature was recorded according to its type, publication date, and the country of the study. Pre-defined key themes were used to analyse the findings of each publication. This was done to categorize the studies into the relevant predefined thematic areas within ageing organizations from a global perspective, as shown below.

### 3. Results

This section is organized in accordance with the predefined themes, as follows: Section 3.1, managing older professionals (*n* = 12); Section 3.2, recruitment and training of older professionals (*n* = 11); Section 3.3, intergenerational relations in organizations (*n* = 8); and Section 3.4, ageism in the workplace (*n* = 21).

### 3.1. Managing Older Professionals

Table 1 below illustrates the chosen studies encompassed in this section (*n* = 12), displaying the author(s) and publication year of each study along with the corresponding journal in which it was published. Additionally, it includes information about the country in which the study was carried out or whether it is a literature review.

**Table 1.** List of selected publications on "managing older professionals".

| Author(s)/Year | Source | Country |
|---|---|---|
| Egdell et al. (2020) [21] | Ageing & Society | Scotland |
| Stengård et al. (2021) [22] | European Journal of Ageing | Sweden |
| Liff and Wikström (2020) [23] | Ageing & Society | Sweden |
| Arman et al. (2021) [24] | Ageing & Society | Sweden |
| Kooij et al. (2022) [25] | European Journal of Work and Organizational Psychology | Netherlands |
| Fasbender and Gerpott (2021) [26] | European Journal of Work and Organizational Psychology | Germany |
| Reed and Thomas (2021) [27] | Management Learning | Literature |
| North (2022) [28] | Frontiers in Psychology | China, US |
| Manzi et al. (2021) [29] | Frontiers in Psychology | Italy |
| WHO Global Report on Ageism (2021) [1] | Report | Global |
| Berger (2021) [6] | Book Chapter | Canada |
| Woolever (2021) [20] | Book Chapter | US |

A United Nations' projection shows that nearly one third of the working force in industrialized nations will be 50 or over by 2050 [20]. As much as governments have been forced to rethink social policy and retirement structures, private organizations will have no option but to redesign their human resources policy framework if they wish to adapt to the current social demographic changes occurring in work-related contexts. In a society where organizations hold a growing influence, it is crucial to not only offer older professionals job opportunities aligned with their skills and profiles but also to acknowledge their work engagement and satisfaction through HR policies and standards [23]. The primary focus should be on creating work environments that condemn ageist attitudes and refrain from stereotyping older professionals as ineffective based solely on their age [1,6,20]. Scholars strongly recommend collaborative efforts from government and organizations that support older professionals [1,6,25–28].

In Scotland, despite endeavours to stimulate the involvement of older adults in the job market through legislation, there is a lack of comprehensive strategies to prepare for the shifts in demographics and the labour market. The assessment of employees' age and their suitability for work tends to be approached on an individual level rather than being addressed in a systematic and structured manner, indicating a somewhat "hands-off approach" from management [21]. Conversely, in Sweden, older workers are perceived as either exemplifying productive ageing and offering a solution to labour scarcity or as a hindrance to the recruitment of younger, more promising employees with innovative skills [24]. Within Swedish workplaces, human resources departments are actively working to uphold narratives concerning the timing of retirement. Nevertheless, in the majority of cases, organizations tend to avoid conversations about extending the retirement age [24].

Regarding management practices within organizations, the concept of "management-by-generation" has gained prominence [27]. This increasingly prevalent managerial approach involves organizing the workforce according to different generations, enabling management to find optimal alignments between employees' characteristics and the tasks at hand. These "generational characteristics" can inform the most appropriate leadership styles and reward systems. The following passage, extracted from a research participant (senior HR manager), provides insight into the implementation of "management-by-generation" [27] (p. 53).

> *"Basically the thinking behind it is there's no good or bad employees, there's differences in the profiles. So we help managers have an understanding of what appeals to different profiles in the workforce. The message is let's acknowledge those differences and their diversity and what appeals to them, and let's help you manage them"*. (Senior HR)

In this way, "management-by-generation" borrows elements from job crafting, adding the age factor to it. Job crafting relates to one's sensibility to both change in and the specificities of a given task or process. It consists of employee self-initiated modifications either in the task or social interaction that improve performance [25]. Another study examines the significance of job crafting, highlighting its pivotal role in retaining older employees by means of "opportunity-enhancing HR practices." Drawing from data from a survey involving 125 older workers (65+) from the Netherlands, the findings reveal that these HR practices are linked to shifts in psychological empowerment, leading to modifications in "utilization crafting behavior" (using personal resources to offset the loss of other personal resources) and "developmental crafting behaviors" (leveraging personal resources for outstanding performance). However, alterations in empowerment did not translate into changes in "accommodative crafting" (managing the reduction of personal resources). Although job crafting naturally evolves and emerges, organizations can establish favourable conditions to foster its growth [25].

Academics advocate the formulation of novel HR policies to acknowledge and enrich employees' experiences, which would not only boost productivity but also foster authenticity, organizational identification, and self-assessed performance. Addressing age diversity can involve concentrating on organizational practices and skills, minimizing the centrality of age. As a result, stereotypical perceptions about older workers would lose significance within the organizational context [29]. However, concerning initiatives related to job crafting, research indicates that HR "accommodation practices" amplify the connection between occupational self-efficacy and knowledge sharing. Yet, the outcomes of such practices are not consistently positive. While HR practices assist employees with greater occupational self-efficacy, they concurrently lead to reduced knowledge sharing among those with lower occupational self-efficacy due to perceived age-based discrimination [26]. In today's organizational landscape, it is imperative for managers to adopt dual roles as both "compliance" and "ethics officers", as ageism is not only morally objectionable but also, in numerous industrialized countries, legally prohibited [20].

Turning to Sweden, researchers investigate the reasons for people retiring earlier than anticipated, despite the existence of policies designed to extend the retirement age. Findings reveal that essential and rational HR procedures, such as development feedback

sessions, salary negotiations, and internal recruitment, influence this trend. However, prioritizing the salary demands of younger employees and reducing training efforts for older workers are perceived as "normal" practices. A viable solution for organizations aiming to extend the working lives of older employees is seen in the redesign of HR policies [23]. Additionally, noticing the scarcity of studies examining the impact of psychosocial working conditions on the timing of retirement among older workers, another Swedish study explores whether suitable psychosocial working conditions could contribute to prolonged working careers [22]. The researchers examined seven waves (2006–2018) of the Swedish Longitudinal Occupational Survey of Health ($n = 6000$, observations = 10,632). Their conclusion suggests that enabling older workers to have control over their tasks, providing avenues for skill utilization and learning, along with recognizing and rewarding their performance, could facilitate the extension of their working lives. Moreover, job resources gain increasing significance with age [22].

### 3.2. Recruitment and Training of Older Professionals

Table 2 below illustrates the chosen studies encompassed in this section ($n = 11$), displaying the author(s) and publication year of each study along with the corresponding journal in which it was published. Additionally, it includes information about the country in which the study was carried out or whether it is a literature review.

**Table 2.** List of selected publications on the "recruitment & training of older professionals".

| Author(s)/Year | Source | Country |
| --- | --- | --- |
| Previtali and Spedale (2021) [30] | Journal of Aging studies | Italy |
| Garthe and Hasselhorn (2021) [31] | Ageing & Society | Germany |
| Laliberte Rudman and Aldrich (2021) [32] | Journal of Aging studies | US |
| Halvorsen et al. (2022) [33] | Research on Aging | US |
| Neumark (2022) [34] | Journal of Aging & Social Policy | US |
| Helleseter et al. (2020) [35] | Journal of Human Resources | Mexico, China |
| Hebl et al. (2020) [36] | Annual Review of Organizational Psychology and Organizational Behavior. | Literature |
| Cebola et al. (2021) [4] | Ageing & Society | Literature |
| WHO Global Report on Ageism 2021 [1] | Report | Global |
| Berger (2021) [6] | Book | Canada |
| Ciampa and Chernesky (2021) [19] | Book Chapter | US |

Drawing from longitudinal data collected from 2835 participants often referred to as "baby-boomers", one study offers substantial evidence concerning occupational shifts within the older workforce in Germany. The study shows that changes among older workers are prevalent (13.4% changed employer, 10.5% changed profession, and 45.1% changed work tasks). Furthermore, the desire for change was unfulfilled for 17.6% regarding profession, 13.2% regarding employer, and 8.9% regarding work. This highlights the necessity for further investigation into the phenomenon of occupational change among older workers, particularly for those who aspire to change but perceive it as an unattainable objective [31]. Indeed, age-based discrimination during the hiring process constitutes a significant barrier for older workers attempting to either join or re-enter the job market [1,4,6,34,36]. Numerous employment policies and practices inadvertently contain age-related biases related to recruitment, selection, performance assessment, and access to training opportunities [19].

A Canadian study underscores the prevalence of negative age-related stereotypes that manifest themselves in recruitment and training prospects, overshadowing positive attitudes toward older employees. Older adults seeking employment reported encountering age-related biases during the hiring process, leading to the development of age-related

stigmas. Employers examined participants' resumes in a biased manner, for instance, selecting candidates for interviews based on the year of their degree completion or the number of years of experience they possessed. Additionally, participants noted that employers seemed to estimate their age during job interviews and used "ageist language during the hiring process" [6] (p. 73). Faced with discrimination from prospective employers, the study participants devised a range of strategies to mitigate the impact of ageism on employment opportunities. These strategies included keeping up with training, particularly in IT-related fields, concealing their age by omitting prior work experiences from their resumes, making modifications to appear younger (e.g., using hair dye), and lowering expectations and employment aspirations—a perceived "realistic" employment potential as described by some employers [32]. In the Chinese and Mexican contexts, employers explicitly incorporate age and gender requirements in specific job requests, particularly in less-skilled positions. The manner in which employers make such demands differs based on task-specific distinctions in the "relative productivity of men and women", indicating that continuous skill enhancement can mitigate the "negative skill-targeting relationship" in these economies [35] (p. 428).

Instigating a transformation in organizational culture stands out as the most effective strategy to combat ageism within workplaces. This approach fosters an environment in which prejudiced age-related stereotypes are suppressed, enabling the establishment of workplace policies grounded in principles of equality and inclusion [30]. Through an examination of the US Federal Senior Community Service Employment Program, researchers delve into how "on-the-job training" for individuals aged 55 and above positively impacts their engagement and well-being. Emphasizing health and well-being outcomes is identified as a key aspect of enhancing satisfaction and performance among older employees [30]. In this regard, initiatives aimed at fostering intergenerational collaboration within the workplace can bolster not only the well-being of workers but also their overall productivity [33], as the subsequent section elaborates upon.

### 3.3. Intergenerational Relations in Organizations

Table 3 below illustrates the chosen studies encompassed in this section (*n* = 8), displaying the author(s) and publication year of each study along with the corresponding journal in which it was published. Additionally, it includes information about the country in which the study was carried out or whether it is a literature review.

**Table 3.** List of selected publications on "intergenerational relations in organizations".

| Author(s)/Year | Source | Country |
| --- | --- | --- |
| Jarrott and Lee (2022) [37] | Research on Aging | Literature |
| Hsu et al. (2022) [38] | Journal of Aging & Social Policy | Taiwan, South Korea, Japan |
| Fasbender and Gerpott (2021) [26] | European Journal of Work and Organizational Psychology | Germany |
| Fasbender and Drury (2021) [10] | European Journal of work and Organizational Psychology | Germany |
| Yeung et al. (2021) [39] | Frontiers in Psychology | Hong Kong |
| Rožman and Milfelner (2022) [40] | Frontiers in Psychology | Slovenia |
| WHO Global Report on Ageism (2021) [1] | Report | Global |
| Jarrot et al. (2021) [41] | Research on Aging | Literature |

Promoting intergenerational relationships and fostering "mutual understanding" can serve as effective tools for combatting ageism and age-based discrimination in the workplace, while also contributing to improved performance [10,38]. Research has demonstrated that targeted intergenerational leadership yields a positive impact on the work engagement of older employees. A study involving 583 older employees in Slovenia emphasizes the significance of addressing the specific needs of older workers through the implementation of age-specific management techniques [40], a notion highlighted in the literature [26,27].

Additionally, this study reveals that the negative effects of intergenerational leadership on emotional burnout are more pronounced in larger organizations compared to smaller ones.

Despite acknowledging the benefits of intergenerational knowledge sharing for organizational performance, German scholars delve into the reasons why this process often encounters obstacles within organizations [10,26]. Based on a sample of over 400 older workers, the study finds that age discrimination is perceived by these individuals as a hindrance to their job performance and capabilities, leading to reduced interaction with younger colleagues [26]. Yet in Germany, scholars focus on age-diverse friendships within the workplace and their implications for the organization [10]. Results show that age-diverse friendships may cultivate a sense of "oneness" among both younger and older employees (*n* = 186). Such a perception brings forth positive outcomes, such as enhanced cooperation, which in turn results in slightly increased job satisfaction as well as in reduced turnover and "interrole conflict" [10]. In Hong Kong, older workers manage their emotional reactions to intergenerational conflicts within organizations by reducing their attention to negative stimuli [39].

Researchers emphasize the necessity for practitioners in intergenerational programs to have access to evaluation, interprofessional collaboration, and programming resources [37]. However, there exists a general challenge in identifying evidence-based intergenerational practices due to limitations in research methodology. Specifically, "intergenerational program research frequently consists of small samples and pre-post analyses of attitudinal data with little attention to implementation characteristics" [41] (p. 283).

### *3.4. Ageism in the Workplace*

Table 4 below illustrates the chosen studies encompassed in this section (*n* = 21), displaying the author(s) and publication year of each study along with the corresponding journal in which it was published. Additionally, it includes information about the country in which the study was carried out or whether it is a literature review.

**Table 4.** List of selected publications on "ageism in the workplace".

| Author(s)/Year | Source | Country |
|---|---|---|
| Previtali and Spedale (2021) [30] | Journal of Aging Studies | Italy |
| Sugisawa (2022) [9] | Ageing & Society | Japan |
| Cebola et al. (2021) [4] | Ageing & Society | Literature |
| Taylor and Earl (2021) [11] | Ageing & Society | Australia |
| Van der Horst and Vickerstaff (2021) [42] | Ageing & Society | Literature |
| Kleissner and Jahn (2020a) [7] | Frontiers in Psychology | Germany |
| Kleissner and Jahn (2020b) [8] | Research on Aging | Germany |
| Axelrad (2021) [43] | Journal of Aging & Social Policy | Israel |
| Kim et al. (2021) [44] | Journal of Aging & Social Policy | 15 OECD countries |
| Fasbender and Gerpott (2021) [26] | European Journal of Work and Organizational Psychology | Germany |
| Goecke and Kunze (2020) [45] | European Journal of Work and Organizational Psychology | US |
| Hebl et al. (2020) [36] | Annual Review of Organizational Psychology and Organizational Behavior | Literature |
| Kreiner et al. (2022) [46] | Annual Review of Organizational Psychology and Organizational Behavior | Literature |
| Reed and Thomas (2021) [27] | Management Learning | Literature |
| Crozier and Woolnough (2020) [47] | Management Learning | England |
| North (2022) [28] | Frontiers in Psychology | China, US |

Table 4. *Cont.*

| Author(s)/Year | Source | Country |
|---|---|---|
| Manzi et al. (2021) [29] | Frontiers in Psychology | Italy |
| WHO Global Report on Ageism (2021) [1] | Report | Global |
| Berger (2021) [6] | Book | Canada |
| Blackstone (2021) [18] | Book Chapter | US |
| Ciampa and Chernesky (2021) [19] | Book Chapter | US |

Using the Comparative Macro-Level Ageism Index, researchers conduct a comparative analysis of ageism across 15 OECD countries [44]. Turkey had the highest ageism score, while Japan had the lowest due to its favourable conditions for older adults' economic and health well-being, along with social engagement. In terms of the workplace, South Korea was identified as the country least likely to practice discrimination, although it still scored high in terms of discrimination against older adults based on their economic status.

Ageism is recognized as a "pan-cultural problem" [28]. Building upon the World Health Organization's perspective that age is a dynamic concept rather than purely chronological [1], scholars characterize subjective age as an ever-evolving social construct that varies "between- and within-person" over time [45]. Currently, considering that older workers frequently encounter discrimination in the workplace [1,6,9,19,26,28,36,43,46], a phenomenon termed "fake age advocacy" has surfaced in Australia. This term refers to efforts that obstruct an informed public discourse on the employment of older workers. To foster a responsible discussion on this issue, the authors propose the reduction of "age norming" in job roles, outlining "five underlying principles" essential for facilitating constructive discourse around older employees, as shown below [11] (p. 01):

*"(. . .) countering myths concerning the extent and nature of age barriers in the labour market; avoiding and challenging the use of age stereotypes in making the business case for older workers' employment; recognition that age interacts in complex ways with a range of other factors in determining people's experiences of the labour market; challenging public understanding that is grounded in the notion that generational conflict is inevitable; and discarding traditional notions of the lifecourse in order to overcome disjunctions and contradictions that hamper efforts to encourage and support longer working lives".*

Through a comprehensive analysis of 33 quantitative and 21 qualitative studies, a systematic literature review focus on work-related ageism underscores the intricate and pervasive nature of this form of discrimination [4]. Ageism manifests across various work domains, encompassing fields like IT, advertising, finance, education, healthcare, journalism, hospitality, employment tribunals, and human resources. It becomes evident in numerous dimensions, including hindrances during the hiring process, challenges related to employability, and the assessment of older workers' performance [4]. Although older employees may experience mental and physical repercussions due to ageism, organizational approaches like "age management" and "management by generation" [27] remain underutilized. Implementing intergenerational initiatives is recognized as effective in addressing ageism, leading to a shift in the overall perception of older workers away from being deemed inefficient [4].

In Italy, a study involving over 8000 participants investigates the combined impact of "age-based" and "gender stereotype threat" on the work identity processes and work performance of older adults [29]. The findings reveal that ageism detrimentally affects the authenticity of older workers as well as their sense of belonging within the organization. For women over 50, ageism and gender stereotypes are twice as likely to manifest in the workplace compared to men in the same age group (ibid). Yet in Italy, scholars discuss age as a significant "ordering" and "divisive" element within the organizational landscape. This concept shapes hierarchies, career trajectories, and the dynamics of the employer–employee

relationship [30]. By analysing video recordings of performance appraisal interviews in a labour union, and grounded in the notion of age as a "constructed social category", researchers identify three modes of "doing age": "quantification" (e.g., years in the organization), "ageing within the organization", and "age-group membership identification" (e.g., 'young' versus 'old') (ibid). The authors argue that work-related ageism undermines the vision of a modern, inclusive workplace, highlighting how the adaptable social context serves as the backdrop for enacting ageism. The study suggests that such ageism is perpetuated through the practices of both employers and employees, operating within a particular organizational culture that facilitates the functioning of the organization.

Echoing the previously mentioned research [29,30], other scholars observe that ageism contributes to discrimination and diminished productivity within the German context [7,8]. The outcomes of the Implicit Association Test reveal "a stable, moderate implicitly measurable preference for younger over older workers" [7] (p. 01). Nonetheless, while younger workers were positively assessed in terms of performance and adaptability, older workers received high scores for competence, reliability, and warmth. Another study conducted in Germany examines the core components of age-related work stereotypes (performance, adaptability, reliability, and warmth) and their variations across a group of 180 nurses aged between 19 and 63. The findings indicate that older nurses were perceived as "more competent, less physically strong, and less adaptable", whereas younger nurses were characterized as "less reliable and less warm." Additionally, a phenomenon known as "in-group bolstering" was identified across all age groups, with particular prominence among older professionals [8] (p. 126).

Academics have observed how the intricacies of one's cultural and organizational context influence stigmatization and discrimination. Stigma can arise from workplace-related factors such as occupation and status, or from personal characteristics like age and disability [46]. On the contrary, employers' ageist attitudes toward older workers stem from the "negative attributes socially attached to older persons as a homogenous group" [30] (p. 01) rather than being rooted in the nuances of the work environment. Furthermore, an examination of the history, present state, and future of contemporary discrimination within US organizations imparts three significant insights: (1) the population of older workers is consistently growing as people extend their working years prior to retirement, (2) despite evidence of the advantages older workers bring to organizations, they are often perceived as less productive, more resistant to change, and financially burdensome, and (3) negative stereotypes concerning older workers are deeply ingrained, and many encounter age-based discrimination at their workplace or during the recruitment process [36]. At work, participants in Blackstone's study [18], comprising American older adults, reported instances of offensive age-related jokes (targeted at themselves and others), "comments or behaviors that demeaned participants' age; and unwanted questions about participants' private lives" [18] (p. 37). Additionally, participants experienced being "isolated from important work activities" (ibid).

The examination of ageism brings to light its multidirectional nature [1]. Research highlights how ageism experienced by young academics within universities contributes to the development of internal conflicts between their self-perception and external expectations of identity. These, in turn, lead to imposter syndrome and a sense of marginalization within the workplace [47]. A proposition is put forth to redefine the term "ageism" to differentiate it from "ableism" (discrimination in favour of able-bodied individuals). Despite the closely intertwined nature of these terms, establishing such a distinction would enable the formulation of more precise and effective policies to provide support for older professionals [42].

## 4. Discussion

In this section, we offer recommendations based on the findings of the systematic literature review, drawing from select publications [11,19,23,27,29,30] and building upon the framework established by the American Association of Retired Persons (AARP) [19] as well as Woolever's ethics framework [20].

The recommendations below revolve around the overarching concept of organizational culture and its impact on employees' well-being and productivity [30]. Given the inevitable ageing of the workforce, the importance of developing intergenerational structures within organizations and implementing age-friendly policies becomes apparent. It is therefore essential to debunk the misconception that older workers are more expensive and less productive for organizations [19]. Many employment policies and practices inherently carry biases based on age, affecting areas such as recruitment, selection, performance assessment, and access to training opportunities. Hence, there is a genuine need to reform unjust organizational policies. As Ciampa and Chernesky put it, "it is ageism, rather than labor cost and performance considerations, that is the reason corporations force out older workers" [19] (p. 95).

What

Redesign and implement policies and regulations to

- Accommodate the needs of older professionals;
- Support and nurture their work performance, intrinsic motivation, and physical and psychological well-being;
- Eradicate preconceptions, stereotypes, and discrimination against them;
- Foster engagement with older professionals, avoiding their premature exclusion from the organization;
- Promote a sense of intergenerational unity within the organization;
- Eliminate the practice of the "age norming of jobs".

How

- Introduce phased retirement options, allowing for a gradual reduction of working hours until retirement;
- Provide flexible work schedules, catering to part-time or full-time arrangements and diverse hour allocations;
- Offer remote work opportunities ("flexlocation");
- Facilitate intergenerational interaction, encouraging older professionals to mentor or coach younger colleagues, enhancing their own well-being and sense of purpose while increasing commitment and productivity;
- Enable job crafting, where HR departments align older professionals with roles that resonate with their expertise and innovative thinking, igniting their intrinsic motivation for greater innovation and productivity;
- Establish conflict resolution mechanisms;
- Ensure consistent feedback on performance;
- Champion visible leadership concerning issues linked to an ageing workforce;
- Deliver high-quality training for all employees, including targeted professional development for older professionals when suitable;
- Adapt the physical work environment to cater to the needs of older workers (e.g., ergonomic structures, proper lighting and acoustics);
- Construct an ethics framework, fully communicating it throughout the organization. Clearly outline organizational values and the consequences of deviating from them. Ethical core values may guide all aspects of human resource activities, from goal setting and resource allocation to communication dissemination, performance evaluation, and job promotion. Incorporate ethics training and education as an integral part of employees' professional development, with the ethics framework undergoing ongoing management review.

While conducting a thorough analysis of each organizational context is key for establishing a high-performance intergenerational work environment, it is possible to outline strategies that can be applied across various professional settings. The foundational step in this process is the establishment of a robust organizational ethics framework, as outlined previously. Over the past couple of decades, several companies, including General Dynamics, Mercedes Benz, BMW, Martin Marietta, Southwest Airlines, Starbucks, and

Hewlett-Packard, have attempted to implement various integrity strategies with varying levels of success [18,19]. Research underscores that the key to an ethical organization lies in viewing ethics as a core value rather than a peripheral aspect of organizational life. While it would be unrealistic to consider the organizational code of ethics as a "holy grail", curing all forms of unethical behaviour, ethics frameworks provide a platform through which organizations can manage unethical attitudes and behaviours, ensuring equitable and responsible conduct [20].

Employers' ageist attitudes toward older professionals stem from the "negative attributes socially attached to older individuals as a homogeneous group" [30] (p. 01). The systematic literature review on ageing organizations reveals two converging yet subtly distinct approaches to shaping a fair and productive workplace: one presented by Manzi et al. [29], and the other highlighted by Reed and Thomas [27]. According to Manzi et al. [29], the evaluation of workers should hinge on their skills, competencies, motivation, and performance, rather than their age. On the other hand, Reed and Thomas [27] emphasize "management-by-generation" as a constructive managerial tool for fostering inclusive organizations. While acknowledging the advantages of aligning employees' characteristics with job specifications through practices like job crafting, we, the authors, lean towards Manzi's et al. [29] approach. This stance also resonates with Taylor and Earl [11] and Woolever [20], aiming to diminish the significance of "age barriers" or "age norming". Such a shift reduces the influence of age when it comes to social interactions, allowing other vital organizational concerns to receive heightened management attention. In this way, stereotypical perceptions of older workers gradually lose relevance within the organizational landscape and culture. However, achieving this goal is undoubtedly challenging due to the widespread occurrence of ageism across cognitive, emotional, and behavioural dimensions globally [1]. Simultaneously, organizations themselves are ageing, underscoring the need for an equitable work environment where productivity is unrelated to one's age, and age-based discrimination is reprimanded. This review shows that a strategic blend of inclusive intergenerational approaches within organizations can yield optimal management outcomes [10,11,19,20,27,29,30]. Possible future research venues within the context of ageing organizations include further investigations into organizational norms and policies that optimize older professionals' skills. More studies that inform optimal job crafting practices and facilitate a comprehensive cultural change that embraces inclusive modes of management, recruitment, and training, are crucial for today's inevitably intergenerational workplace.

**Author Contributions:** Conceptualization, T.C., A.S.F. and T.T.; methodology, T.T., A.S.F. and T.C.; formal analysis, A.S.F. and T.T.; investigation, T.T. and A.S.F.; resources, T.T. and A.S.F.; data curation, A.S.F. and T.T.; writing—original draft preparation, A.S.F.; writing—review and editing, A.S.F. and T.T.; visualization, A.S.F.; supervision, A.S.F., T.T. and T.C.; project administration, A.S.F., T.T. and T.C.; funding acquisition, T.C. All authors have read and agreed to the published version of the manuscript.

**Funding:** This research was funded by the Irish Research Council, New Foundations Scheme 2022 (project number P61375).

**Institutional Review Board Statement:** Not applicable.

**Informed Consent Statement:** Not applicable.

**Data Availability Statement:** The data are unavailable due to privacy or ethical restrictions.

**Conflicts of Interest:** The authors declare no conflict of interest.

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
