# Peer review of "Ageing Organizations: Reviewing the Literature and Making a Few Recommendations for Human Resource Management"

_merits, doi:10.3390/merits3040038_

Round 1
Reviewer 1 Report
Comments and Suggestions for Authors
The format here is to be a be a review articles categories the existing literature on aging workers in organization in the following areas:
Managing older professionals
Recruitment and training of older professionals
Intergenerational relations in organizations
Ageism in the workplace
I feel that the following points could also be given some attention with a reference or two:
identifying the 4 generations that are represented in the workplace of today
some of the possible reasons for agism in the workplace which might be appearance, physical decline, and challenges or unfamiliarity with technology being generalized to other areas
identifying objective strengths of older workers which emanate from experience and wisdom and somehow crafting job descriptions and responsibilities in ways that would capitalize on those strengths
Comments on the Quality of English Language
The language looks OK. I assume that "ageism" is an aceptable alternate spelling for what I am familiar with (agism)
Author Response
Dear Reviewer,
Thank you very much for your review. We have worked on it entirely, as follows:
- The introduction now contextualizes the study more comprehensively. We have addressed the suggested points (some of the possible reasons for agism in the workplace; and identifying strengths of older workers) – please see lines 36–44. Regarding the four generations that are represented in the workplace today, however, we have reviewed the literature review results and found no direct mention to the four generations, as suggested – and well observed by the American Management Association. In this case, which is unfortunate, we would not be able to add this information to the manuscript. We would like to thank you for your understanding on this matter.
- The conclusion has been also enhanced, now illuminating future research venues – please see lines 507–12.
Please see the revised manuscript attached.
With many thanks and kind regards,
The authors
Reviewer 2 Report
Comments and Suggestions for Authors
The paper addresses an interesting topic and one that is potentially relevant to the readers of Merits. The paper presents a systematic literature review of 30 recent studies on ageing organizations and the associated management challenges in four thematic areas aiming to make recommendations for human resource management.
My general impression of the paper is positive and broadly speaking, the stated objective is fulfilled. The review provides a good picture of the findings from prior work in each one of the four thematic areas.
What seems to be missing is the identification of ‘what is there to be uncovered’ in this field, the topic areas in which more research is needed, and what the author(s) believe to be the most promising research avenues in each one of the four thematic areas. Briefly, the paper would gain with some suggestions on the key issues that require further research on managing older professionals, their recruitment and training, intergenerational relationships, and work-related ageism.
Some minor issues:
- (line 170), a reference is missing
- (lines 300), it is not clear what (=186) means
Author Response
Dear Reviewer,
Thank you very much for your review. We have addressed it entirely, as follows:
- We have identified the topic areas in which more research is needed, and what we believe to be the most promising research avenues regarding the thematic areas. We have added it to the conclusion section. Please see lines 507–12.
- (line 170), a reference is missing. The reference has been added (now in lines 166 and 169).
- (lines 300), it is not clear what (=186) means. Corrected (now in line 304).
The revised manuscript follows attached.
With many thanks and kind regards,
The authors